# Preparation, Characterization, and Application of Sodium Alginate/ε-Polylysine Layer-by-Layer Self-Assembled Edible Film

Ruohan Bao [1], Xingfen He [2], Yifan Liu [2], Yuecheng Meng [2] and Jie Chen [2],*

1   Taizhou Vocational College of Science & Technology, Taizhou 318020, China
2   School of Food Science and Biotechnology, Zhejiang Gongshang University, Hangzhou 310018, China
*   Correspondence: chenjie@zjgsu.edu.cn; Tel.: +86-13516812119

**Abstract:** The edible film (LBL film) was prepared by layer-by-layer self-assembly technology using Sodium alginate (SA) and ε-polylysine (ε-PL) as polyanion and polycation, respectively. The self-assembly method was optimized, the mechanical and physical properties of the optimal LBL film were characterized, and its preservation effect on blueberry was explored. Results suggested that the transmittance of LBL2 film was above 85%, and the appearance was smooth and transparency consistent. We found that the structure was compact and uniform through scanning electron microscopy (SEM). Differential scanning calorimetry (DSC) showed that LBL2 film had excellent thermal stability. And had lower water solubility and inhibitory effect on *Alternaria*. LBL2 films were significantly effective at inhibiting the growth of mold and maintain the firmness of postharvest blueberries. Compared with the control group, LBL2 films could prolong the shelf life of a blueberry by 1~2 days at room temperature.

**Keywords:** sodium alginate; ε-polylysine; layer-by-layer self-assembly; edible film

## 1. Introduction

Society and academia have become increasingly interested in edible films with natural and biodegradable functions, as environmental protection and food safety become increasingly important. As a packaging material for food, fruit, and vegetable, an edible film has broad application prospects. As a biological macromolecular substance, the edible film comprises polysaccharides, proteins, and lipids with edible supplements such as plasticizers through the interaction of electrostatic force and intermolecular hydrogen bond, whose interaction forms a kind of porous mesh structure by using packages such as infiltration, coating, or spray processing. This membrane-like porous mesh structure forms on the surface of fresh fruits and vegetables [1]. The primary function of edible films is to extend the shelf life of fruits and vegetables by controlling the proportion of gas within fruits and vegetables, reducing their respiration rate, and acting as a water moisture-proof layer to delay the dehydration of fruits and vegetables [2–4].

Recently, food waste caused by food quality and safety problems has negatively impacted the country's resources and economic development [5]. Food deterioration caused by oxidation can alter the characteristics of food structure [6]. The edible films and coatings can protect food (milk, fruit, and vegetables) from microbial contamination, extend food shelf life, and improve the mechanical properties and structural integrity of food [7,8]. For example, historically, the fruit waxing technology has long been used to preserve citrus. In recent decades, consumer awareness of edible, biodegradable, and environmentally friendly materials has increased. Edible films—now widely used in the food processing industry—may replace synthetic or plastic packaging materials to protect food and vegetables, such as peach [9], apples [10], grapes [11], mangoes [12], bananas [13],

pomegranates [14], cherries [15], pineapples [16], cantaloupes [17], cucumbers [18], carrots [19], and tomatoes [20].

Layer-by-layer self-assembly (LBL) is a multilayer composite film formed by spontaneous aggregation on a solid carrier through the interaction force of groups in solution (including electrostatic force, hydrogen bond, chemical bond, and coordination bond) by using the principle of alternating adsorption layer-by-layer [21]. In most cases, the LBL adsorption process is based on the attraction of anionic and cationic charges located on the molecular structure of the polyelectrolyte [21]. The main feature is that the oppositely charged polyelectrolytes—anions and cations—are alternately adsorbed on the surface of the charged substrate through electrostatic interaction. LBL has recently emerged as a popular new preservative coating technology, with promising application prospects for preserving the freshness of fruits and vegetables. Brasil et al. (2012) studied the effects of β-cyclodextrin/cinnamaldehyde LBL coating on the quality and shelf life of fresh-cut pineapples stored at 4 °C, and found that LBL assembly, with the incorporation of microencapsulated antimicrobial, was effective in extending the shelf life and quality of fresh-cut papaya [22]. Martiñon et al. (2014) found that a multilayered edible coating comprising 2wt.% trans-cinnamaldehyde, 2wt.% chitosan, and 1wt.% pectin could help extend the shelf life of fresh-cut cantaloupe by up to 9 days [23].

Sodium alginate (SA) is a natural anion polysaccharide compound extracted from seaweed or kelp [24], and it is a potential biopolymer film or coating material owing to its properties of thickening, stability, suspension, film formation, gel formation, and stable emulsion [25]. SA is easily soluble in water because of large proportions of carboxyl groups, and it can mix with other substances to optimize the properties of SA film [26]. Shima et al. [27] first identified ε-polylysine (ε-PL) isolated from the medium of *Streptomyces albulus* 346 in 1977. As a natural antiseptic for antimicrobial peptides, ε-PL has the advantages of safety, non-toxicity, good thermal stability, biodegradability, and broad-spectrum antimicrobial activity [28]. However, its application is limited because antimicrobial activity may be weakened if it interacts with anionic ingredients within fruit matrixes. Ge Liming et al. (2022) developed antibacterial dialdehyde sodium alginate/ε-polylysine microspheres (DSA-PL MPs) to effectively prolong the shelf life of fruit. Dialdehyde sodium alginate (DSA) was prepared by periodate oxidation of sodium alginate [29]. This preparation method has many steps and uses many reagents. Therefore, it is not convenient and green enough in the practical application process. The combined effects of a novel ε-polylysine/sodium alginate (PLSA) treatment on the sensory and physicochemical characteristics of Japanese sea bass (Lateolabrax japonicas) were investigated by Cai Luyun et al. (2015) [30]. However, only one treatment group was sodium alginate coating combined with 0.4% ε-polylysine. They did not study the coatings with layer-by-layer self-assembly (LBL). Nevertheless, the application and mechanism of ε-polylysine/sodium alginate LBL coating on the maintenance of blueberry quality is still rare and unclear so far.

In order to increase flexibility, improve physical properties, and antibacterial activity of edible films, the main objective of our study was to produce edible films using layer-by-layer self-assembly technology combining SA with ε-PL. And the mechanical and physical properties of the LBL films were characterized. Then, compared with treatments, whether LBL2 films could prolong the shelf life of a blueberry.

## 2. Materials and Methods

### 2.1. Materials

SA was supplied by Qingdao Bright Moon Algae Group Co., Ltd. (Qingdao, Shandong, China), and ε-PL by Zhejiang New Yinxiang Biological Engineering Co., Ltd. (Taizhou, Zhejiang, China). Molecular weight 0.8–2.0 kDa. Glycerin, a plasticizer, was obtained from Shanghai Lingfeng Chemical Reagent Co., Ltd. (Shanghai, China). The glass substrate was cleaned with distilled water. *Alternaria* (CGMCC 3.17853) was purchased from China General Microbiological Culture Collection Center. Potato dextrose agar (PDA) and Bengal red medium were purchased from Hangzhou Microbial Reagent Co., Ltd. (Hangzhou,

Zhejiang, China). Fresh 'Britewell' blueberries (*Menditoo × Tifblue*) were harvested in the Wangjiajing blueberry picking garden, Zhuji city of Zhejiang Province, China (120°13′30″ E, 29°38′1″ N).

## 2.2. Film Preparation

The glass substrates were first immersed in a solution of negatively charged 1.5wt.% SA for 2 min before being washed three times with deionized water. The washing step was to remove excess SA. The substrates were then immersed in positively charged ε-PL (4wt.%) solution for 2 min and washed to remove excess ε-PL. The preceding steps were repeated until the desired number of SA/ε-PL layers were attained. All five SA/ε-PL treated samples were completed with the ε-PL deposition as the last treatment step, as shown in Figure 1. These samples were labeled as LBLn, where n indicates the number of SA/ε-PL layers. Before each test, the substrate strips were conditioned for 48 h at 25 °C and 50% relative humidity (RH) [31].

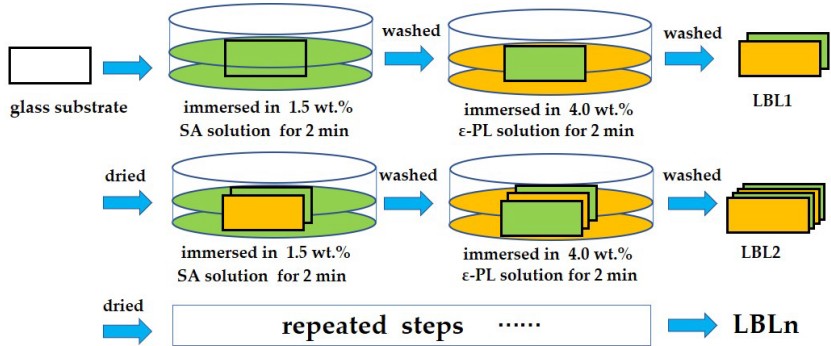

**Figure 1.** Schematics of film preparation.

## 2.3. Thickness and Transmittance Measurements

Film thickness was measured at five points through a digital micrometer (accuracy of 0.001 mm) (Links, Harbin, China). The film thickness was then used to determine the mechanical properties of the films. According to the method of Zhong et al. [32], the transmittance (T) of the film was studied by visible spectrophotometer at 520 nm wavelength.

## 2.4. Mechanical Properties

A texture analyzer (TA-XT plus, Stable Micro Systems, Godalming, UK) was used to measure the mechanical properties of the film samples, including tensile strength (TS; MPa) and elongation at break (EAB; %), according to a previous method [33]. All film samples were cut into rectangles with a length of 40 mm and a width of 10 mm. The initial grip separation was set to 20 mm with a tensile speed of 10 mm/s and an effective stretching separation of 20 mm. Each film type was replicated nine times. TS (MPa) was calculated by dividing the maximum load force at the break by the cross-sectional area. EAB (%) was calculated as a ratio of the elongation at the point of sample rupture to the initial length of a sample.

## 2.5. Water Vapor Permeability (WVP)

The WVP of the films was determined gravimetrically according to the method of Jiang et al. [34]. The uniformly-ground anhydrous calcium chloride was placed in an oven at 105 °C and dried for 2 h. Pre-dried anhydrous calcium chloride (3.0 g) was used as a desiccant in an uncovered centrifuge tube (10 mL). The films were adhered to the tops of the weighing bottles and tightly fixed with double-faced adhesive tape pasted on the rim of the bottles. The weighing bottles were then placed in a chamber with constant temperature (25 °C) and humidity (50% RH). The weights of the uncovered centrifuge

tubes were measured every 12 h for 5 days. The WVP of each sample was estimated using the following equation:

$$\text{WVP}\left(\text{gma Pa}^{-1}\text{s}^{-1}\text{m}^{-2}\right) = \frac{\text{F} \times \text{e}}{\text{A} \times \text{S} \times (\text{RH}_1 - \text{RH}_2) \times 3600 \times 12}$$

where WVP is the water vapor permeability (g m Pa$^{-1}$ s$^{-1}$ m$^{-2}$), F is the slope of the linear weight graph, A is the surface area exposed (m$^2$), e is the average thickness of the film (m), S is the saturation pressure at 25 °C (Pa), and (RH$_1$ − RH$_2$) is the difference in RH between the two sides of the film.

### 2.6. Color Difference Analysis

The sample was cut into a square of 10 mm × 10 mm, placed on an A4 paper printed with the letter 'z', and photographed to compare its transparency. Specific values were measured by a colorimeter. $\Delta E$ was calculated as the following equation:

$$\Delta E_{ab}^* = \left[(\Delta L)^2 + (\Delta a)^2 + (\Delta b)^2\right]^{1/2}$$

### 2.7. Water Solubility

Each film was cut into square pieces (20 mm × 20 mm) and vacuum-dried in a 55 °C oven for 48 h, allowing sufficient time to achieve constant weight before measuring the initial dry weight. Subsequently, the films were immersed in 30 mL of distilled water at 25 °C with gentle agitation. After 24 h of immersion, the specimens were recovered from distilled water and dried to constant weight at 55 °C. Three replicate experiments were conducted, and the result is reported as the average of three values. The solubility is expressed using the following equation [35]:

$$\text{Water Solubility (\%)} = (\text{M}_{\text{start}} - \text{M}_{\text{final}})/\text{M}_{\text{start}} \times 100 \tag{1}$$

### 2.8. Fourier Transform Infrared Spectroscopy (FT-IR)

The structure of the films was analyzed using an FT-IR spectroscope (Thermo Fisher, Pittsburgh, PA, USA) that scanned from 4000 to 400 cm$^{-1}$. To remove the interference of water, we freeze-dried the films in advance.

### 2.9. Differential Scanning Calorimetry (DSC)

Differential scanning calorimetry (TG−DSC STA449F3, NETZSCH, Selb, Germany) was performed in the range of 25–350 °C with a 20 mg sample under a continuous flow of dry nitrogen gas (10 °C/min) at a heating rate of 10 °C/min [35].

### 2.10. Scanning Electron Microscopy (SEM)

The SEM images of the SA and LBL films were observed using a scanning electron microscope (SU1510, Hitachi, Tokyo, Japan). Before using the scanning electron microscope, each sample was gold plated with a vacuum-sputtering coater. The samples were observed with an acceleration voltage of 5 kV.

### 2.11. X-ray Diffraction (XRD)

The crystallinity index of composite films was analyzed using X-ray diffraction (XRD) (XRD-7000, Shimadzu, Tokyo, Japan) in the 2θ range of 5–50° at 2°/min as described in a previous study [36].

### 2.12. Antimicrobial Property

Referring to the method of Buonocore et al. [37] with minor modifications, the prepared suspension of *Alternaria* was added to the PDA medium at a ratio of 1:100 and mixed well. The medium–suspension mix was poured into a 20-mL sterile culture dish until the medium

solidified for reserve use. Perforators were used to cut the film into a 15-mm-diameter circle, and the film was affixed to the surface of the solidified culture medium containing bacterial suspension. Because LBL2 films rolled up quickly after absorbing water, the tweezers were sterilized using an alcohol lamp; once cooled, the membrane was pressed gently against the surface of the medium around the membrane using tweezers, allowing it to fit closely with the smooth surface medium. After 2 h of diffusion at 4 °C (refrigerator), the bacteriostatic ring diameter was measured at 28 °C under culture conditions. LBL2 films were balanced for more than 48 h in a constant temperature and humidity chamber at 25 °C and 50% RH. No substances were added to the CK group. The diameter of the bacteriostatic circle was measured using the cross method, and the average value of the three repeated experiments was considered.

### 2.13. Blueberry Coating Treatment

We refer to the method of Yan et al. [38] with minor modifications, as shown in Figure 2. CK group: control group blueberries were soaked in sterile water for 2 min and then dried. SA treatment group: blueberries were soaked in 1.5wt.% SA membrane solution containing 0.5wt.% glycerol for 2 min, then removed and dried. ε-PL treatment group: blueberries were soaked in 4wt.% ε-PL solution for 2 min and then taken out and dried. LBL2 treatment group: blueberries were immersed in 1.5wt.% SA film solution for 2 min, taken out and dried, and then immersed in 4wt.% ε-PL solution for 2 min, taken out and dried. That was a layer of self-assembled film, repeated once, and the blueberry's surface can be covered with LBL2 film. The above groups of blueberries were dried and stored in baskets at 25 °C and 50% RH incubators. The corresponding physical and chemical indexes and physiological and biochemical indexes were measured every day. At the same time, approximately 50 g of blueberries were taken from each group and stored at −80 °C after liquid nitrogen treatment.

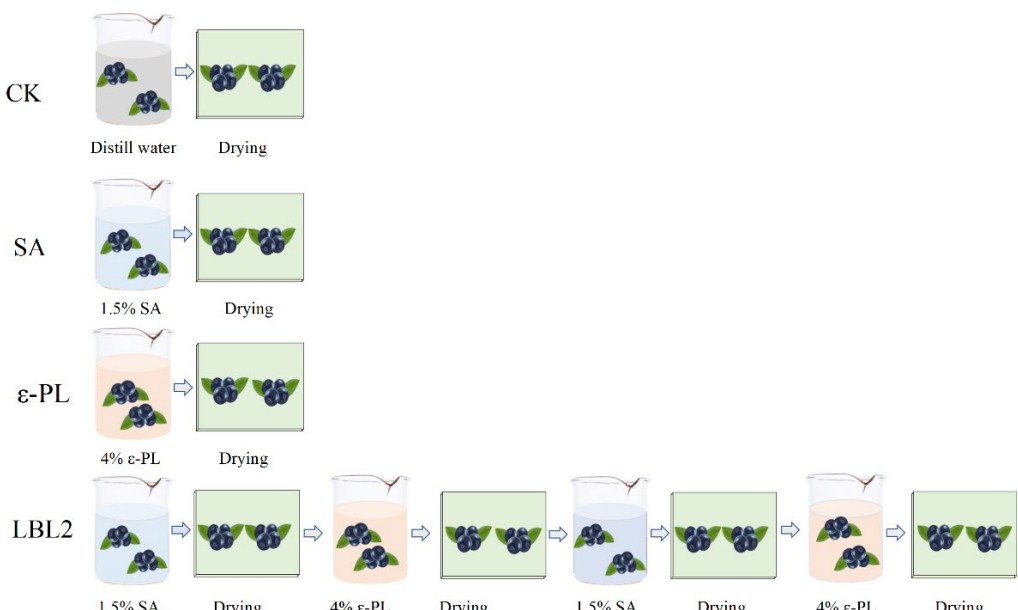

**Figure 2.** Schematics of CK, SA, ε-PL, and LBL2 treatments.

### 2.14. Determination of Decay Rate, Firmness and Total Number of Molds

The number of rotten fruits was counted with sterile gloves. The decay rate (%) refers to the proportion of rotten fruits in the total. Rotten fruit refers to the juice outflow, skin softening, and rotting.

The fruit firmness was measured by texture analyzer. The P/5 stainless steel probe (diameter 5 mm) was selected, the downward pressure distance was 5.0 mm, and the measurement speed was 1.0 mm/s. The peak height of the first peak was the maximum

force, which was used to represent the firmness value. The results were expressed as g and 10 fruits were randomly selected for each measurement. After removing the maximum and minimum values of each group, the average value was taken.

According to GB 4789.15-2016, the number of molds on the fruit's surface was detected and counted by a plate pouring method. In each experimental group, 10 g of blueberry was taken and placed in a sterile homogenizing bag containing 90 mL of normal saline. The homogenizer was beaten for 2 min to make a 1:10 sample homogenate. The sample solution was diluted to the appropriate concentration and counted in Bengal red medium.

### 2.15. Data Analysis

All statistical results were analyzed using three replicates in a completely randomized design. Statistical differences were performed by one-way analysis of variance (ANOVA). Data were analyzed statistically by repeated measures using SPSS 23; $p < 0.05$ was considered statistically significant by Duncan's multiple range test.

## 3. Results and Discussion

### 3.1. Tensile Strength and Elongation at Break

The mechanical properties (TS and EAB) of LBL1, LBL2, LBL3, LBL4, and LBL5 films were compared. As shown in Figure 3, with the increase in the number of SA/ε-PL film layers, the TS of films presented a trend of continuous increase, whereas the EAB of films presented a trend of increasing initially and then decreasing, as the film thickness had a significant influence on the mechanical properties of films, ranged between 56.7 and 87.4 MPa. With the increase in the number of coating layers, the thicknesses of films grew, which enhanced the mechanical properties of the films; consequently, the TS increased gradually. However, film layers were correlated with smaller intramolecular forces between the SA/ε-PL chains. This resulted in a decrease in polymer matrix mobility which was a barrier to film elongation, which is consistent with Adzaly et al. (2015) who reported that the film had much lower (6%) but had a much higher TS (41.4 MPa) [39].

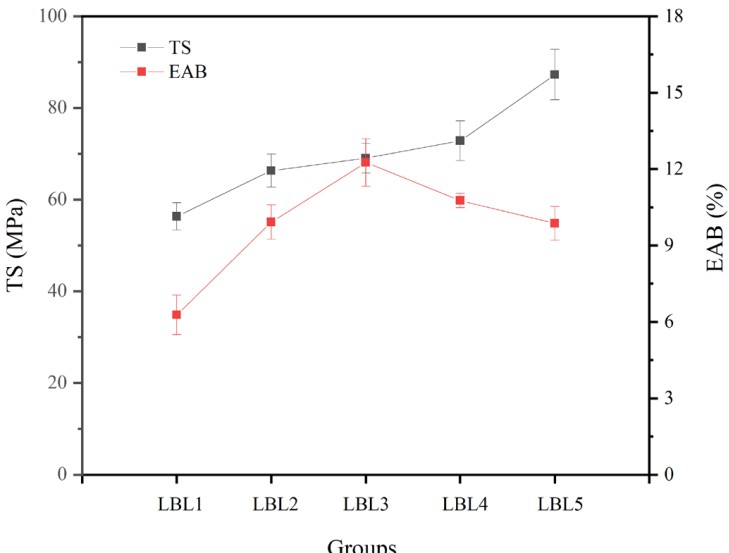

**Figure 3.** Effect of multilayer on tensile strength (TS) and elongation at break (EAB) of the films.

### 3.2. Water Vapor Permeability and Transmittance

The water vapor barrier properties of LBL1, LBL2, LBL3, LBL4, and LBL5 films were compared. As shown in Figure 4, with the increase in the number of film layers, the WVP and transmittance of films gradually decreased. Because of electrostatic adsorption and accumulation, the cross-linking between SA as a polyanion and ε-PL as a polycation on the surface of films was closer. And its structures were also denser, causing the WVP

to decrease. The decrease in transmittance of films may be due to the increase in film thickness of LBL films. But the transmittance of LBL2 (Figure 4) film was above 85%, higher than LBL3, LBL4, and LBL5, but there were no significant differences compared with LBL1 ($p > 0.05$), indicating that the SA and $\varepsilon$-PL molecules were compatible due to decreasing starch granules and more homogeneous matrices that limited diffusion of water vapor. Maltol plasticized the matrices which interacted with starch via H-bonding [40] (Promhuad et al., 2022).

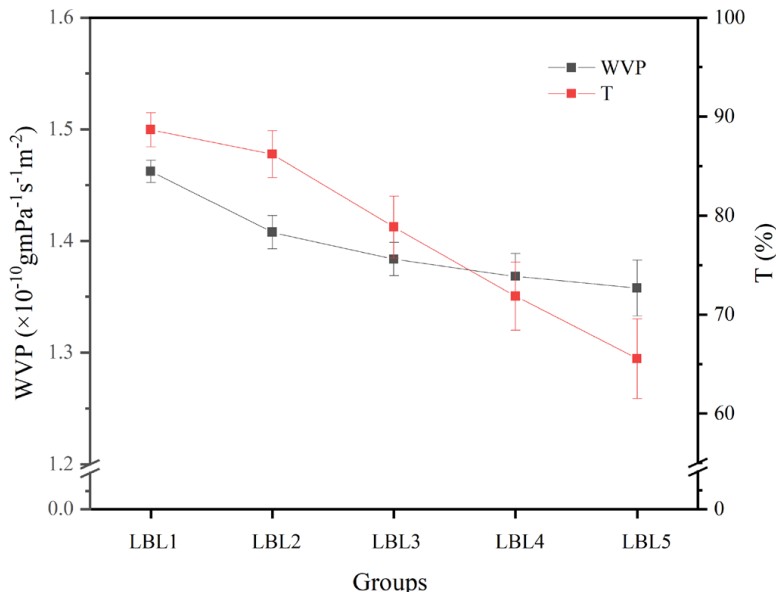

**Figure 4.** Effect of multilayer on water vapor permeability (WVP) and transmittance (T) of the films.

### 3.3. Appearance and Color Difference Analysis

As shown in Figure 5, the appearances of SA film and LBL2 film are smooth, and the transparencies are consistent. The values of L, a and b had no significant difference between SA and LBL2 film, respectively ($p > 0.05$, Figure 6). The results of this experiment showed that LBL2 film could still have good light transmittance after adding an appropriate amount of $\varepsilon$-PL, which also confirmed the reliability of the LBL2 film preparation process.

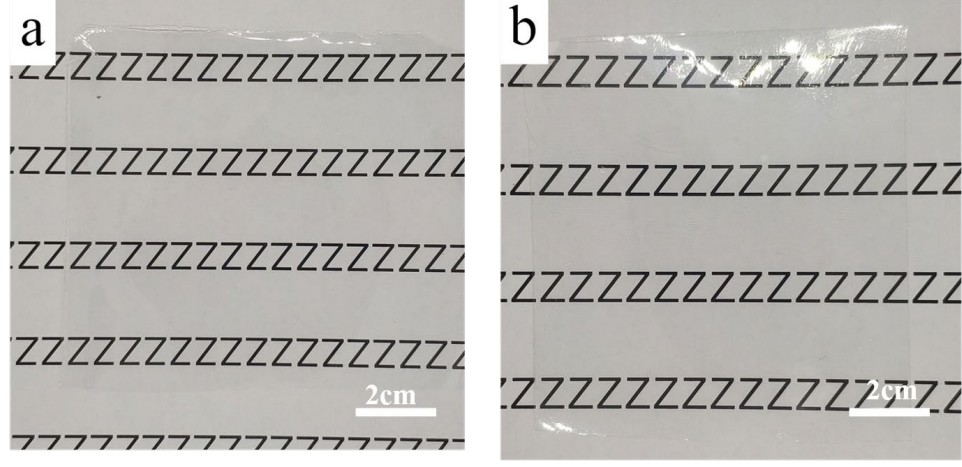

**Figure 5.** Appearance images of SA film and LBL2 film ((**a**): SA film; (**b**): LBL2 film).

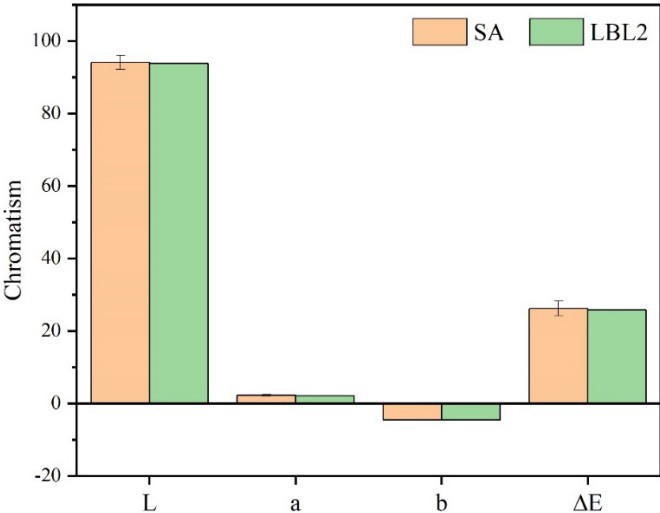

**Figure 6.** Chromatism of SA film and LBL2 film.

### 3.4. Solubility

Solubility is an index of the stability of packaging films in the presence of solvent molecules on the food surface. If the packaging film is soluble, it will deteriorate the food's quality. The water solubility value of pure SA film was 100% (Figure 7) because of carboxyl and hydroxyl groups in SA, which increase the binding with water molecules, resulting in a highly soluble membrane. We found that the water solubility of LBL2 film decreased significantly (from 100% to 7.38%) with an increase in SA/$\varepsilon$-PL. Because the carboxyl group in the SA molecule bound to the amino group on the $\varepsilon$-PL molecule, the arrangement tended to be standardized and dense, and the intermolecular force became larger, which limited the fracture of the molecular chain and the penetration of solute molecules [41] (Rhim, 2004). To reduce the water solubility of SA, we must broaden the packaging applications of LBL2.

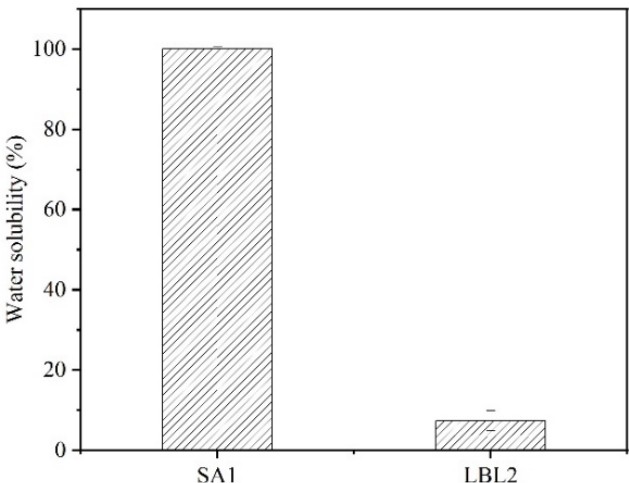

**Figure 7.** Water solubility of pure sodium alginate (SA) and LBL2 films.

### 3.5. Fourier Transform Infrared Spectroscopy (FT-IR)

As is evident from the FT-IR spectrum of $\varepsilon$-PL (Figure 8), the stretching vibration of N–H of primary and tertiary amine was observed at 3449 cm$^{-1}$ and 3253 cm$^{-1}$, respectively. The peak at 2936 cm$^{-1}$ corresponds to the stretching vibration of CH$_2$ [42]. The peaks at 1665 cm$^{-1}$ and 1560 cm$^{-1}$ correspond to the vibration of amide I and amide II [43]. The

peaks at 1257 cm$^{-1}$, 1160 cm$^{-1}$, and 1106 cm$^{-1}$ correspond to the stretching vibration of C-N of aliphatic primary amines [42].

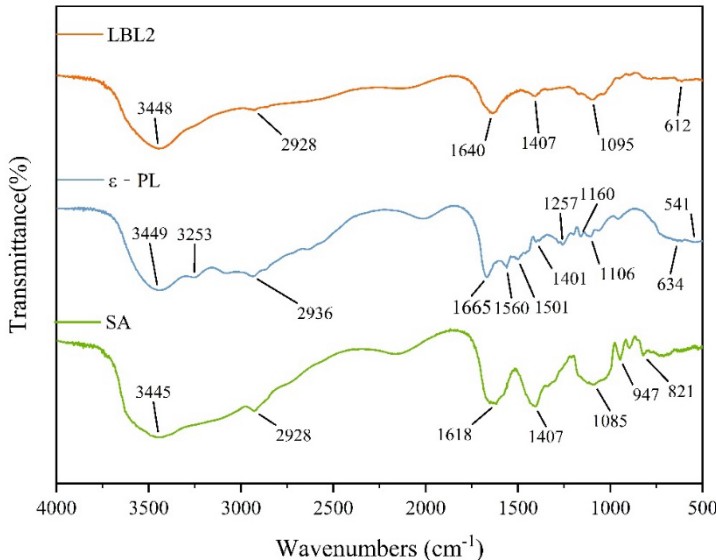

**Figure 8.** FT-IR of pure sodium alginate (SA), ε-polylysine (ε-PL), and LBL2 films.

From the FT-IR spectrum of SA (Figure 8), three bands appear at 3600–1600 cm$^{-1}$: a broad band centered at 3445 cm$^{-1}$ corresponds to hydrogen-bonded O–H stretching vibrations [44], a band at 2928 cm$^{-1}$ corresponds to C–H stretching vibrations [45], and asymmetric and symmetric stretching of carboxylate O-C-O vibration appeared at 1618 cm$^{-1}$ and 1411 cm$^{-1}$, because these were formed from the mannituronic acid of SA [46]. The stretching absorption peak of C-O-C in pyranoid compounds was at 1085 cm$^{-1}$, the stretching vibration peak of C-O in the uronic acid residues was at 947 cm$^{-1}$, and the characteristic peak of mannituronic acid residues was at 821 cm$^{-1}$ [44]. The results are consistent with that of a previous study [47], which reported that the characteristic peaks of alginate appeared at 3386, 1604, and 1411 cm$^{-1}$, corresponding to hydroxyl (OH), carbonyl (C=O), and carboxyl (COOH), respectively.

As shown in Figure 8, the stretching vibration of C-OH at 821 cm$^{-1}$ in LBL2 film was slightly weaker than that in pure SA film, which may be due to the cross-linking between the amino group and the hydroxyl group in the SA [48]. With the addition of ε-PL in LBL2 film, the stretching vibration peak of C=O shifted and migrated from 1618 cm$^{-1}$ to 1640 cm$^{-1}$. The amino group in the ε-PL and the hydroxyl group in the SA molecules could be cross-linked, thereby forming a strong interaction between the molecules of the composite membrane, which verified the improved mechanical properties and thermal stability of the composite membrane.

### 3.6. Differential Scanning Calorimetry (DSC)

Biopolymers are subjected to temperature-dependent structural changes during their applications. We compared the thermal behavior between LBL2 film and pure SA film. The thermal stability of SA composites was studied through DSC, as shown in Figure 9. Both SA and LBL2 films exhibited heat absorption peaks at approximately 110 °C, which were caused by the evaporation and detachment of water that was not removed during the drying process of the film at this stage [49]. Both pure SA and LBL2 films had a characteristic exothermic peak at 200–250 °C because SA decomposed into stable intermediate products, accompanied by the fracture of glycoside bonds, and adjacent hydroxyl groups removed water molecules [50]. The higher the peak temperature, the higher the thermal stability of the film. The characteristic exothermic peak of SA film appeared at 205.3 °C, while that of LBL2 film appeared at 225.9 °C, which is 5.16% higher than that of pure SA film. ε-PL

exhibited good thermal stability and decomposed at 250 °C [51]. After self-assembly with SA, the thermal stability of LBL2 film was improved, and the stability of LBL2 film was enhanced, further indicating a specific interaction between SA and ε-PL molecules.

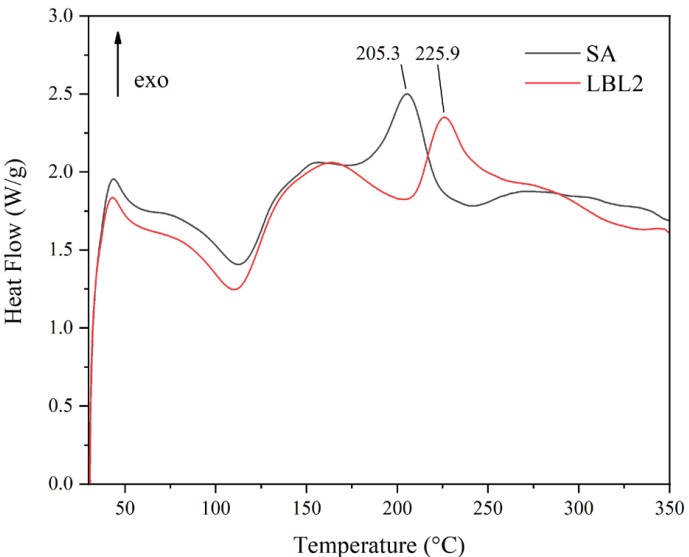

**Figure 9.** DSC of pure sodium alginate (SA) and LBL2 films.

### 3.7. Scanning Electron Microscopy (SEM)

SEM images can be used to analyze the microscopic morphology of pure SA and LBL2 films, which can explain the coating's smoothness and thickness from a microscopic perspective. Figure 10 shows that the film surface of pure SA was relatively flat and smooth, whereas the fracture surface of SA was relatively flat, indicating that SA has excellent film forming properties. The film surface of LBL2 was rough, with many high and low undulations. However, when observed from the film fracture surface of LBL2, there was no apparent void structure, and the structure was relatively compact, further indicating that SA was well integrated with ε-PL. The cross-sectional examination of the film fracture surface of LBL2 revealed a layered structure, indicating that LBL was successful. The thickness of LBL2 film was greater than that of pure SA film, indicating that SA was an anion and ε-PL was a cation bind in an orderly manner.

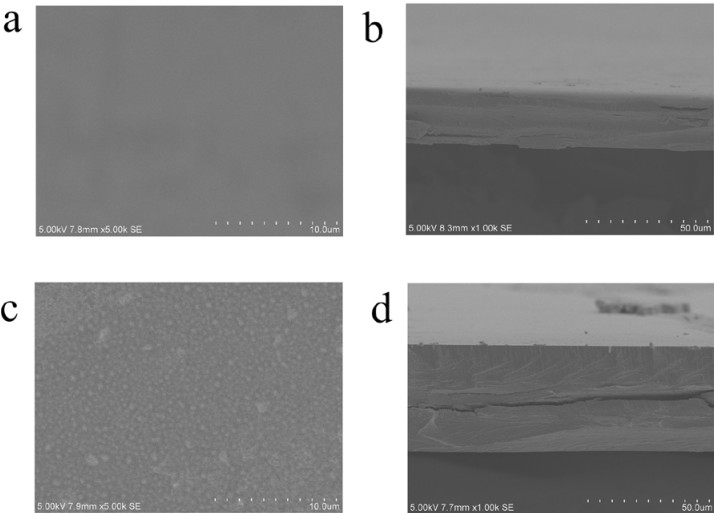

**Figure 10.** SEM images of (**a**) surface of pure sodium alginate (SA), (**b**) fracture surface of pure sodium alginate (SA), (**c**) surface of LBL2 film, and (**d**) fracture surface of LBL2 film.

### 3.8. X-ray Diffraction (XRD)

XRD is an essential method for analyzing the crystalline state of materials at the molecular level and studying the properties and structures of crystalline materials. According to Figure 11, the pure SA film had a strong diffraction peak between 2θ = 19.0° and 2θ = 21.0°. The characteristic peaks of LBL2 films disappeared and were transformed into a broad diffraction peak. These results indicated that LBL2 films have amorphous structures with no crystallizations, and also confirmed the strong interaction and good compatibility between SA and ε-PL in SA/ε-PL self-assembled edible films.

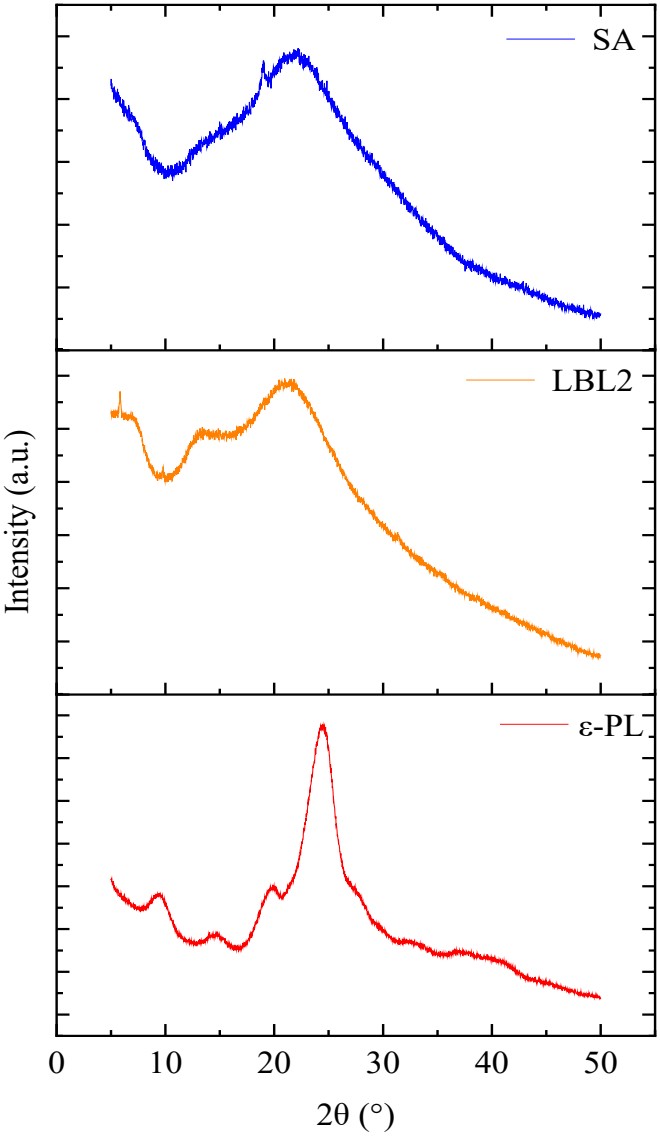

**Figure 11.** XRD of pure sodium alginate (SA), pure ε-polylysine (ε-PL), and LBL2 films.

### 3.9. Antimicrobial Property

The diameter of the inhibitory zones can directly reflect the bacteriostatic effect. As shown in Figure 12, inhibitory zones were observed around the LBL2 film with a diameter of 4.0 ± 0.2 cm, which was larger than CK. As we all know, *Alternaria* is a common pathogen in blueberry. The inhibitory mechanism of ε-PL may be due to the inhibitory ability of ε-PL on the growth and virulence of mold, stimulating the accumulation of intracellular ROS, reducing the expression of the PR gene in pathogens, damaging the cytoplasm membrane

of mold, and inducing the expression of NADPH oxidase homologous gene in a respiratory burst [52].

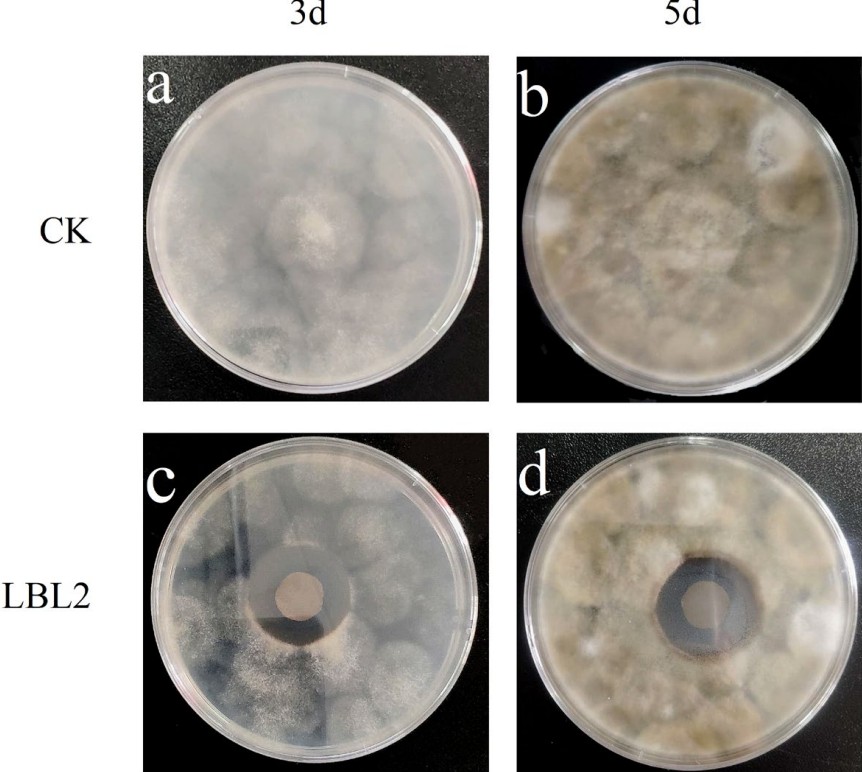

**Figure 12.** Inhibitory zones against *Alternaria*. (**a**) Day 3 inoculation in CK group, (**b**) Day 5 inoculation in CK group, (**c**) Day 3 inoculation in LBL2 group, (**d**) Day 5 inoculation in LBL2 group.

*3.10. Effects of Different Treatments on Decay Rate, Firmness, Total Mold Number and Appearance of Blueberry*

During storage, the decay rate of blueberry showed an upward trend. The comparison between groups showed that LBL2 < ε-PL < SA < CK, and there was a significant difference on the 2nd day ($p < 0.05$). ε-PL had a good broad-spectrum antimicrobial effect, and SA could also isolate external microorganisms after film formation. Therefore, the ε-PL and SA groups were not significant in the first 3 days; the spoilage rates were 25.33% and 23.39%, respectively. However, during the later storage period, there may be original colonies on the surface of blueberry, which leads to skin corruption. The ε-PL group was significantly lower than the SA group ($p < 0.05$). However, the LBL2 group always performed better, the decay rate is the lowest. This has a certain correlation with the firmness of the fruit, which is one of the important indicators reflecting the storage resistance of the fruit. Excessive softening is a key factor leading to the decay of blueberry.

As shown in the Figure 13, during the storage, it can be seen from the comparison between groups, especially in the later storage period, the LBL2 group maintained significantly higher firmness ($p < 0.05$). On the 5th day, the firmness of CK group was 174.38 g, the firmness of SA group, ε-PL group, and LBL2 group were 273.02 g, 220.24 g, and 383.76 g, respectively. On the first day, the firmness of the SA and LBL2 groups clearly increased, which indicated that the coating increased the firmness of blueberry, which was consistent with the results of Duan et al. [53]. Layer-by-layer self-assembly coating treatment can effectively delay the decrease of blueberry fruit firmness, which is consistent with the results reported in the literature that the coating maintains low pectinase and cellulase activities during the storage of Hami melon [54].

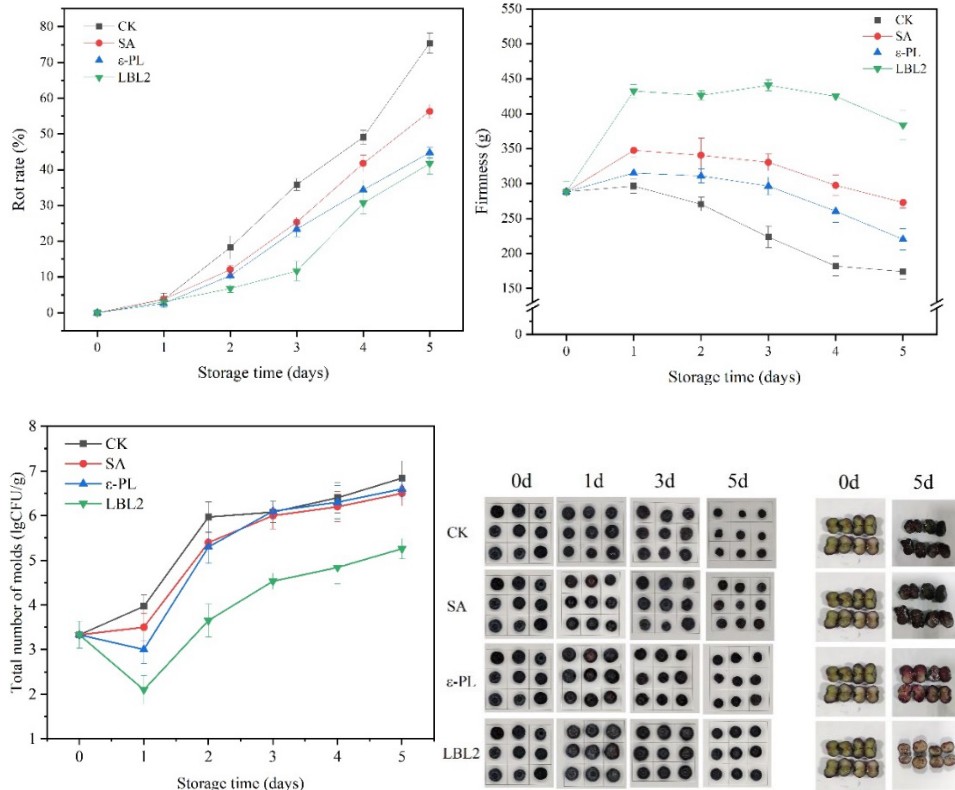

**Figure 13.** The change in rot rate (%), firmness, total number of molds (log CFU/g), and appearance of the blueberry during storage. CK indicates the control; SA indicates 1.5% Sodium alginate coating; ε-PL indicates 4% ε-polylysine coating; LBL2 indicates layer-by-layer coating. Vertical bars indicate standard deviations.

At the same time, blueberry fruits are susceptible to spoilage mold contamination during storage. As shown in the Figure 13, the total number of mold colonies in the CK group showed a continuous upward trend. Comparison between groups showed that the total number of mold colonies in the LBL2 group was the lowest. On the 3rd day, the total number of mold colonies in LBL2 group was $3.65 \pm 0.37$ lg CFU/g, which was significantly lower than that in ε-PL group, SA group, and CK group ($p < 0.05$). The results showed that the main pathogenic strains causing postharvest decay of blueberry were *Alternaria*, *Penicillium*, and *Botrytis cinerea*. The inhibitory effect of LBL2 treatment on *Alternaria* was also obvious, as shown in Figure 12. The initial colony number of blueberries treated with LBL coating decreased, which inhibited the growth and reproduction rate of mold and delayed the decay of the fruit. ε-PL may stimulate the accumulation of reactive oxygen species in mold cells, damaging the cytoplasmic membrane of mold, thereby inhibiting the growth and reproduction of mold [52]. Antimicrobial polymers for several foods are a key trend in edible and biodegradable packaging which reduces food loss and waste, and provides sustainability in food industry [55,56].

Sensory quality is one of the most intuitive indicators to measure the quality of fruit and vegetable storage. As shown in the Figure 13, during the storage period, the blueberry in the CK group lost more water, shrunk seriously, the flesh color became darker, and the flesh rotted. After SA, ε-PL, LBL treatment in the first 2 days of storage had good sensory quality. On the 3rd day, the blueberry in the CK group began to shrink, the fruit became soft, and some blueberries showed mold growth, while the LBL2 treatment group had relatively full fruit, good firmness, and quality. SA delayed the decrease of blueberry firmness, titratable acid and surface brightness, and had a certain effect on the preservation of blueberry [57]. ε-PL can also delay the decline of blueberry quality. However, with the extension of storage time, the effect of SA and ε-PL decreased. The respiration of fruit was

enhanced, and the quality of blueberry decreased significantly. LBL2 not only inhibited the respiration of its fruits and microorganisms, but also maintained the antibacterial activity of ε-PL, which maintained better blueberry quality.

## 4. Conclusions

This study investigated the effects of different self-assembled layers on the basic properties (mechanical properties, WVP, and transmittance) of LBL films; an LBL2 film with good basic properties was selected for characterization. FT-IR spectroscopy and SEM measurements indicated that SA and ε-PL were successfully LBL self-assembled by electrostatic stacking. The XRD and DSC results further proved the compatibility of LBL2 films. Compared with pure SA, LBL2 film had higher thermal stability, better antimicrobial properties, lower moisture content, and water solubility, broadening the membrane's practical applications. In the future, we should research the primary and secondary metabolisms as well as improve the quality of blueberry senescence during storage. Overall, LBL films could be a potential approach to maintain the quality of post-harvest blueberry and extend shelf-life.

**Author Contributions:** Conceptualization, J.C.; data curation, R.B. and Y.L.; formal analysis, R.B.; funding acquisition, J.C.; methodology, X.H. and R.B.; project administration, Y.M.; resources, J.C.; software, R.B.; supervision, Y.M. and J.C.; validation, R.B.; writing—original draft, R.B. and J.C. All authors have read and agreed to the published version of the manuscript.

**Funding:** This research was funded by Key research and development program of Zhejiang Province (Grant No. 2019C02088), National Natural Science Foundation of China (Grant No. 31871830), and Natural Science Foundation of Zhejiang Province (Grant No. LY18C200002).

**Institutional Review Board Statement:** Not applicable.

**Informed Consent Statement:** Not applicable.

**Data Availability Statement:** Not applicable.

**Conflicts of Interest:** The authors declare no financial or other conflicts of interest in this work.

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
