# Peer review of "Preparation, Characterization, and Application of Sodium Alginate/ε-Polylysine Layer-by-Layer Self-Assembled Edible Film"

_coatings, doi:10.3390/coatings13030516_

Round 1

Reviewer 1 Report

This article with the title " Preparation, characterization, and application of sodium alginate/ε-polylysine layer-by-layer self-assembled edible film" is useful for the food industry

1-         The introduction section still lacks to present significant information and it is not comprehensive.

2-      Authors are required to perform further significant analysis, for example, zeta potential for LBL1, LBL2, ...

3-      Some discussions of results need to be discussed with further details and compared with previous literature to show the differences, for example, tensible section, why TS increased continuously and EAB increased and then decreased? (Explain the reason by citing reliable articles)

4-      In the solubility section, why does É›-pL decrease solubility? Explain the reasons and compare them with other similar articles

5-      Authors added some details but they missed to report and justify the changes in the glass transition temperature (Tg) from DSC peaks.

6-      Some discussions of results need to be discussed with further detail and compared with previous literature to show the differences.  

7-      Figure quality is poor, Improve the quality of the figures; especially figure 12  (pictures of Blueberry during shelf life)

Reviewer 2 Report

L79 Objective should be clearly stated in this paragraph

Fig. 5 add statistical analysis

L252 Add more discussion e.g., Starch is hydrophilic materials and the bulky modified groups also enhanced water absorption and diffusion which increased water affinity (Promhuad et al. Polymers14(24), 5342)

L322 are there small peak?

Fig. 12 Picture of fruit is too small to see the detail.

L378 there is no evidence on respiration. Should use term “likely” or “possibly”.

L395 Add more discussion e.g., Antimicrobial polymers for several foods is a key trend in edible and biodegradable packaging which reduces food loss and food waste and provides sustainability in food industry (San et al. Polymers14(18), 3793 ; Srisa et al. Polymers14(19), 4042)

Reviewer 3 Report

I reviewed the research article entitled: “Preparation,characterization and application of sodium alginate/ε-polylysine layer-by-layer self-assembled edible film”. 

I found the research interesting and with potential for practical food applications. 

Here are some comments and curiosities I have about the manuscript.

Abstract 

1.     Line 11: it might be better to use “smooth” instead of “flat”. This change should be made throughout all the manuscript (line 170, 236, 303).

2.     Line 12: reference should be made to the meaning of LBL2.

3.     Line 15: amino group. 

Introduction

I found the introduction to be adequate, with numerous references and well-defined research objectives.

Materials and methods

1.     Line 86: please, explain the culture medium of Alternaria CGMCC 3.17853. 

2.     Line 87: what other chemical compounds are you referring to?

3.     Line 98: please add the total number of layers analyse in this research or at least, the number of layers of the films that were submitted to the different characterization experiments. 

4.     Line 105: between which wavelengths was the transmittance studied?

5.     Line 172: why only LBL2 films were submitted to the antimicrobial test?

6.     Line 173: which is the CK group?

7.     Line 205: please add the description of Bengal red medium. 

Results and discussion

An improvement of the discussion is necessary in the entire results and discussion section. Discuss the values and results obtained with similar literature and research publications. 

1.     Line 230: why this value is an indicator of good transparency? Please, add reference.

2.     Line 235: at some point of the results and discussion section you should explain at some point that the best results were obtained with LBL2 and the rest of analysis were made only with these films.

3.     Line 247: not only this, if the packaging is very soluble it could not be used to cover food matrices, due to the high moisture of the most part of foods.

4.     Line 328-329: why is this diameter optimal?

5.     Line 337: why was blueberry selected as food matrix model for the assay of LBL films?

Conclusions

You can improve the conclusions section by adding what might be future experiments or what commercial vision such food applications might have.

Reviewer 4 Report

The article relates the preparation, characterization and application of sodium alginate/ε-polylysine films. The manuscript is interesting but was not prepared on desirable professional level. I would like to recommend a major revision of this paper. 

1.       In the introduction, the prior art is incomplete as there are already publications on the alginate/ε-polylysine combination and layer-by-layer self-assembled techniques ( it is worth describing the current state of knowledge about these issues on the basis of the latest bibliography). Consequently, the same applies to references. Also, the authors, after improving the state of the art, should better emphasize what is the novelty of the work in comparison with previous publications on the same type of films.

2.       The description of methodology for obtaining the films is unclear. It is better to provide one scheme to show the experimental process for better understanding.

3.       In section 3.1 and 3.2, LBL3 foil shows better application properties, why the authors chose LBL2 for further research? Please justify.

4.       In section 3.5 L. 277 -280 the authors conclude on the basis of FTIR spectroscopy "The amino group in the ε-PL and the hydroxyl group in the SA molecules could be crosslinked, thereby forming a strong interaction between the molecules of the composite membrane, which verified the improved mechanical properties and thermal stability of the composite membrane", while in the abstract they write "Differential scanning calorimetry (DSC) showed that the carboxyl group of interacted with the amino of ε-PL".

5.       In DSC studies, to enable better interpretation, it would be worth making a thermogram for the ε-PL sample. It would also be worth determining the Parameter of enthalpy of melting DH (J/g), correlating the values with the crystal structure shown in section 3.8.

Overall, this paper presents the data as such rather than a correlation between them, so that I recommend a major revision before publication.

Round 2

Reviewer 1 Report

The authors have done the tests well and have the ability to be published in the journal

Reviewer 4 Report

Please correct the formatting of the text. However, I believe that the scheme of making the foil would be useful to the readers (2.2 Film Preparation).
